

# Identification of operational deflection shapes of a wind turbine gearbox using fiber-optic strain sensors on a serial production end-of-line test bench

Unai Gutierrez Santiago[1,2], Aemilius A.W. van Vondelen[1], Alfredo Fernández Sisón[2], Henk Polinder[1], and Jan-Willem van Wingerden[1]

[1]Delft University of Technology, Faculty of Mechanical Engineering, Mekelweg 2, 2628 CD, Delft, The Netherlands
[2]Siemens Gamesa Renewable Energy, Parque Tecnológico de Bizkaia, 48170 Zamudio, Spain

**Correspondence:** Unai Gutierrez Santiago (u.gutierrezsantiago@tudelft.nl)

**Abstract.**

Wind energy has witnessed a staggering development race, resulting in higher torque density demands for the drivetrain in general and the gearbox in particular. Accurate knowledge of the input torque and suitable models are essential to ensure reliability, but neither of them are currently available in commercial wind turbines. The present study explores how a subspace

identification framework, using distributed fiber-optic strain sensors on a four-stage gearbox, can provide input torque measurements through the use of operational deflection shapes. Compared to conventional gear tooth root strain gauge measurements, an innovative measurement setup with 129 fiber-optic strain sensors has been installed on the outer surface of the ring gears to research the deformations caused by the gear mesh events. Consistent estimates of the deflection shapes have been found by applying the Multivariable Output-Error State-sPace (MOESP) subspace identification method to strain signals measured

on a serial production end-of-line test bench. These operational deflection shapes, driven by periodic excitations, account for almost all the energy in the measured strain signals. Their contribution is controlled by the torque applied to the gearbox. From this contribution, a torque estimate has been derived for dynamic operating conditions. Accurate knowledge of the input torque throughout the entire service life allows future improvements in assessing the remaining useful life of wind turbine gearboxes. Additionally, tracking the operational deflection shapes over time is proposed to enhance condition monitoring in planetary

gear stages.

## 1 Introduction

The growth of wind energy in the last decades has been remarkable. The Global Wind Energy Council (GWEC) reported 93.6 GW of new wind energy capacity installed in 2021, 72.5 GW onshore and 21.1 GW offshore, bringing the worldwide

cumulative wind power capacity to 837 GW (GWEC, 2023). That is a 3.5 times increase in the last 10 years from a global



capacity of 237.7 GW in 2011 and a staggering 35 times increase in 20 years from a total capacity of 23.9 GW in 2001 (IRENA, 2013). The projected market growth for the coming years and decades is even larger. In 2022, the International Renewable Energy Agency (IRENA) and International Energy Agency (IEA) published a road map for the energy sector to become net-zero by 2050, with the aim of limiting the rise in global temperatures to 1.5°C (IEA, 2023). According to this

proposal, the annual capacity additions of wind energy should reach 390 GW by 2030, 310 GW onshore and 80 GW offshore. That means increasing the yearly installations more than four times compared to the ones recorded in 2021 and more than five times compared to the average over the last three years (IEA, 2023).

To facilitate such rapid growth, the main focus of industry and academia has been lowering the levelized cost of energy from wind (LCoE) (van Kuik et al., 2016). This push to lower the LCoE has resulted in a race from wind turbine manufacturers

to increase the rotor diameter, power rating, and hub height of wind turbines. The evolution of size in offshore turbines has been even more dramatic because they have less stringent logistic constraints (Musial et al., 2023). To illustrate the pace of growth, the average values of wind turbines installed during the years 2011, 2016, and 2021 have been summarized in Table 1. Power rating, rotor diameter and hub heights were provided in the yearly cost of energy reviews (Tegen et al., 2012; Stehly et al., 2012; Stehly and Duffy, 2022). The trend to increase power ratings, rotor diameter, and hub heights can be understood

from a wind turbine's fundamental power generation equation (Veers et al., 2019). The power produced by a wind turbine is proportional to the air density, the power coefficient, the cubic exponent of the wind speed and the area swept by the rotor. Increasing hub height reduces the influence of surface friction, allowing wind turbines to operate in higher-quality resource regimes where wind velocities are higher. More power per turbine allows fewer turbine installations, lower balance-of-system costs, and fewer moving parts (for a given power capacity level), thereby enhancing reliability. Larger rotors capture more

energy and a substantial reduction in costs can be achieved because blade lengths can be increased while many other costs remain fixed. In addition, a trend to grow the size of the rotor relative to the generator rating allows for a lower rated wind speed and more frequent operation at full power output, resulting in a higher capacity factor.

If we assume the blade tip speed to remain constant, torque will increase with the cube of the rotor diameter. From the yearly average power rating and rotor diameter values provided by Stehly et al. we have estimated the associated rotor torque using

the maximum tip speed figures provided in the yearly reviews, which are 80 m/s for onshore turbines and 90 m/s for offshore. As it can be seen in Table 1, the rotor torque of the installed turbines has increased more than 3 times onshore and more than 3.3 times offshore in a time period of just 10 years. This rate of development is unprecedented in any other industry or engineering application and ensuring turbine reliability remains a top priority (Veers et al., 2022).

Torque is the main sizing factor for the drivetrain and the gearbox. The drivetrain makes a large contribution to the capital

expenditure of the turbine and also affects other turbine costs because increasing the tower-top mass has a substantial impact on the main frame, tower, and foundation. The pressure to lower costs and the size constraints due to handling and logistic limitations have translated into higher torque density demands for wind turbine gearboxes. The torque density values in Table 1 have been estimated using equivalent gearbox models for such power ratings and hub diameters (Gamesa Gearbox, 2023). The increase in torque density witnessed in just a decade is enormous. Thanks to multiple technological innovations, torque

densities of 200 Nm/kg are now considered state-of-the-art from different gearbox manufacturers (ZF-Wind-Power, 2020;





**Table 1.** Average onshore wind turbine power rating, rotor diameter, and hub height.

|  | Onshore | | | Offshore | | |
| --- | --- | --- | --- | --- | --- | --- |
| **Year** | 2011 | 2016 | 2021 | 2011 | 2016 | 2021 |
| **Power rating (MW)** | 1.5 | 2.2 | 3.0 | 3.6 | 4.71 | 8.0 |
| **Rotor diameter (m)** | 82.5 | 108 | 127 | 107 | 128 | 159 |
| **Hub height (m)** | 80 | 84 | 95 | 90 | 93.7 | 102 |
| **Rotor torque (MNm)** | 0.77 | 1.46 | 2.38 | 2.14 | 3.35 | 7.07 |
| **Torque density (kg/Nm)** | 70 | 100 | 130 | 140 | 150 | 200 |

Power rating, rotor diameter and hub height data from (Tegen et al., 2012), (Stehly et al., 2012),

(Stehly and Duffy, 2022), input torque and torque density estimated by the authors.

Winergy, 2020; Gamesa Gearbox, 2023). For such high torque ratings and torque density values, a trend has emerged in new gearbox architectures towards more planetary stages and more planets per stage. In wind turbines with a power rating of up to 2 MW, the most widely used gearbox architecture comprises a single planetary stage and two parallel gear stages (Oyague, 2011). In the range from around 2 MW to 6 MW, gearboxes with two planetary stages and a single parallel stage have become

mainstream. For higher power ratings, gearboxes with three planetary stages are expected to become dominant. Due to the large number of planets in the input stages and the limitations in outer diameter, the space available for planet bearings has decreased to a point where journal bearings have to be adopted because there is not enough space for roller element bearings.

Overall, gearbox complexity is increasing in the pursuit of lighter designs, while maintaining gearbox reliability is mandatory to ensure low operational expenses. Two key factors are essential to achieve successful designs. On one hand, accurate

knowledge of the loading conditions throughout the complete service life of the gearbox is crucial. On the other, accurate models are required to predict its performance and maintenance requirements. Unfortunately, sensors that provide detailed load measurements of the turbine during commercial operation are not generally available (Dykes et al., 2019). It is possible to estimate the input gearbox torque from the electric currents in the generator but normally, this information is only available through the SCADA and it cannot capture the torque fluctuations caused by the dynamic wind turbine operation, especially in

damaging events like emergency brake events (Egeling et al., 2018).

As a consequence, a direct measurement of the actual torque is needed. The traditional method to measure torque, based on strain gauges on the rotating shaft, is considered impractical for commercial wind turbines due to the expensive nature of the equipment required and not suitable for long-term applications (Perišić et al., 2015). The need for novel sensing technologies and measurement techniques that facilitate a fleet-wide implementation of torque measurements has resulted in great

research interest. Zhang et al. (2018) explored alternative direct measuring techniques and discussed the associated technical and economic difficulties. An alternative direct measurement method, based on deformation measurements on the outer surface of the first stage ring gear of the gearbox, was proposed by the authors of this work (Gutierrez Santiago et al., 2022). Other researchers have focused on indirect techniques or the so-called virtual sensors where a model of the system is combined with data from sensors in other locations of the turbine to obtain estimated data of the input torque (Perišić et al., 2015; Azzam et al.,



2021; Cappelle et al., 2021; Mehlan et al., 2023; Mora et al., 2023). These virtual sensing approaches require accurate wind turbine and drivetrain models but the complexity of current designs exceeds modeling capabilities (Veers et al., 2022). Costly experimental evaluation is needed to achieve the desired degree of confidence (Bucher and Ewins, 2001). Data-driven modeling techniques, also referred to as system identification in the systems and controls community, provide a framework to estimate models of dynamical systems when the accuracy of physical models derived from first principles is unsatisfactory. System identification is well-established in mechanical structures (Al-Khazali and Askari, 2012), where it is more widely referred to as experimental modal analysis (EMA). Either with an instrumented impact tool or a shaker EMA relies on measuring a controlled applied force to identify frequency response functions and modal parameters from the system response. However, in the case of large structures, it is difficult to excite the system with enough energy to produce measurable outputs. Operational modal analysis (OMA) is an alternative output-only approach that overcomes the difficulty of exciting the system by relying on ambient broadband excitation. There is a trend to replace EMA with OMA because in OMA the excitation and boundary conditions of the system are those seen in operation and are deemed more representative of the structure's real use in service (Hermans and van der Auweraer, 1999). In the specific case of rotating machinery like wind turbine gearboxes, several factors impede using OMA (Ozbek et al., 2013). Most notably, the input excitation is unknown and may not adequately excite all modes of interest (Thibault et al., 2012), and the premise of having a white noise excitation in the frequency range of interest is violated because periodical loads due to rotating elements act on the system and typically dominate the system response (Di Lorenzo et al., 2017-01-01). Research interest in overcoming these difficulties has increased recently, and many different algorithms have been proposed. Van Vondelen et al. reviewed the main algorithms used for wind turbines and classified them using nine suitability criteria (van Vondelen et al., 2022). These criteria included the accuracy of the algorithms, the ability to distinguish closely spaced modes, computational complexity, and the ability to handle periodic stationary and non-stationary excitation or harmonics. When structural modes and harmonics are widely separated and when the rotor speed is constant over time, harmonics are identified by the OMA algorithms as artificial modes with zero damping.

The main contributions of this paper are:

- We develop and describe a novel measurement setup for a wind turbine gearbox comprising 129 fiber-optic strain sensors installed and distributed around the ring gears of the three planetary stages, and we present the results of measurements performed in a serial end-of-line test bench.

- We apply the multivariable output-error state space method (MOESP) to identify the periodic modes, referred to as operational deflection shapes, which has enabled quantifying the unknown periodic excitations and has been found to provide an estimation of the input torque of the gearbox.

- For modern gearboxes with high torque density demands, we propose a framework based on accurate input torque knowledge throughout the entire service life to improve the assessment of the consumed fatigue life and tracking the operational (periodic) deflection shapes for fault detection in the planetary stage components of the gearbox.

The remainder of the paper is structured as follows. In Section 2, the chosen identification framework is motivated, and the key definitions and formulation are provided. In Section 3, we describe the measurement setup and the test wind turbine





gearbox together with the experimental conditions. In Section 4, the key findings of using subspace identification on strain
signals are shown, and finally, Section 5 draws the main conclusions of this work, and recommendations are given for future
work.

## 2   Subspace system identification framework

This section describes the theoretical framework used to identify operational deflection shapes from strain data collected by
fiber-optic sensors. Starting from the state-space representation used, we justify how the periodic inputs can be modeled within
the system matrix, leading to a stochastic identification problem. Once the system matrices describing the dynamic behavior
have been estimated, up to a similarity transformation, we show how the estate and output measurements can be reconstructed
using a Kalman filter.

We assume that the system to be identified is a finite- dimensional, linear, time-invariant system, subject to measurement
and/or process noise, that admits a discrete-time innovation state-space representation (Veen et al., 2013) given by:

$$x_{k+1} = Ax_k + Bu_k + Ke_k, \tag{1}$$

$$y_k = Cx_k + Du_k + e_k, \tag{2}$$

where $x_k \in \mathbb{R}^n$, $u_k \in \mathbb{R}^{n_u}$, $e_k \in \mathbb{R}^{n_y}$, $y_k \in \mathbb{R}^{n_y}$, are the state, input, innovation signal and output vectors, respectively. The
ergodic white-noise sequence $e(k)$ is assumed to be uncorrelated with the input sequence $u(k)$. The assumption of a linear
time-invariant system is considered valid when the gearbox operates close to rated torque conditions. Under these operating
conditions, contact patterns in the gear flanks are fully developed, and non-linear effects like backlash or material properties
related to the torque reaction arm elastomers are not expected to play a role. The matrices $A \in \mathbb{R}^{n \times n}$, $B \in \mathbb{R}^{n \times n_u}$, $K \in \mathbb{R}^{n \times n_y}$,
$C \in \mathbb{R}^{n_y \times n}$ and $D \in \mathbb{R}^{n_y \times n_u}$ are the system, control, Kalman gain, sensor, and output matrices, respectively. The system
dimension or order of the system is $n$, and the dimension of the output vector $y_k$ is the number of measured response signals
$n_y$.

Operational modal analysis relies on ambient broadband excitation and assumes this excitation is random white noise in
the frequency range of interest. In this case no deterministic input is considered (i.e., $u_k = 0$), which leads to the so-called
stochastic realization problem. In wind turbine gearboxes, and rotating machinery in general, this premise is severely violated
because the periodic action of shafts and gears dominates the system response. Gres et al. showed that it is possible to extend the
stochastic realization to OMA under (unknown) periodic excitations (Greś et al., 2021) by modeling the effect of a deterministic
periodic force as a sum of a finite number of $h$ sinusoidal frequency components such that $u(t)$ has the shape:

$$u(t) = \sum_{i=1}^{h} a_i \sin(\omega_i t + \phi_i), \tag{3}$$

where $a_i, \omega_i, \phi_i \in \mathbb{R}$ are the periodic input components' unknown amplitude, frequency, and phase. This approach has been
successfully applied to an operational offshore wind turbine and shown to provide accurate estimates of the first three tower
bending modes (van Vondelen et al., 2023). The sinusoidal frequency components can become part of a combined state vector





to eliminate the deterministic input component from Eqs. (1)-(2) and yield an equivalent state space realization without the periodic input as:

$$\bar{x}_{k+1} = \bar{A}\bar{x}_k + \bar{K}e_k, \tag{4}$$

$$y_k = \bar{C}\bar{x}_k + e_k, \tag{5}$$

where the extended system matrix $\bar{A}$ can be rewritten as:

$$\bar{A} = \begin{bmatrix} A^{\mathrm{sys}} & A^{\mathrm{b}} \\ 0 & A^{\mathrm{per}} \end{bmatrix}. \tag{6}$$

This extended system matrix combines the periodic and structural modes which, due to the upper right block structure, can be distinguished because the eigenvalues of the periodic part correspond to undamped modes on the unit circle.

The objective of system identification is to estimate the matrices $\bar{A}$ and $\bar{C}$, up to a similarity transformation, using only the output measurement $y_k$. For the present study, the subspace method Multivariable Output-Error State-sPace (MOESP) was chosen because it has been shown to provide asymptotically unbiased estimates of model parameters as long as the system input has adequate persistency of excitation (Verhaegen and Dewilde, 1992) and the RQ factorization enables a computationally efficient implementation. Furthermore, using instrumental variables it is possible to deal with process and measurement noise. The full description and proofs of the algorithm are given in (Verhaegen and Verdult, 2007) and the implementation shown in this paper was accomplished using the LTI System Identification toolbox (Houtzager, 2012) for Matlab®. The user must define three key parameters when realizing the MOESP algorithm:

1. $N$: the number of samples for each of the signals.

2. $s$: the number of block rows, used to construct the Hankel matrices.

3. $n$: the model order.

The matrices $\bar{A}_T$ and $\bar{C}_T$ are the estimates, up to a similarity transformation of $\bar{A}$ and $\bar{C}$. That is, $\bar{A}_T$ has the same eigenvalues as the matrix $\bar{A}$ and the system $(\bar{A}_T, \bar{C}_T)$ has the same input-output behavior as the original system $(\bar{A}, \bar{C})$. These linear transformations are given by: $T^{-1}\bar{A}T$, $\bar{C}T$ and $T^{-1}\bar{K}$ with $T \in \mathbb{R}^{n \times n}$. The transformed state is such that $\bar{x} = Tx$. With a suitable transformation matrix, it is possible to transform the system $(A_T, C_T)$ into the so-called modal form with a diagonal state-transition form or combine complex-conjugate pole pairs to form a real, "block-diagonal" system in which $\bar{A}_M$ has two-by-two real matrices along its diagonal. The dynamics of the system are completely characterized by the eigenvalues (poles) and the observed parts of the eigenvectors (mode shapes) of the $\bar{A}_M$ matrix. The eigenvalue decomposition of $\bar{A}_M$ is given by:

$$\bar{A}_M = [\Phi][\Lambda][\Phi]^{-1}. \tag{7}$$

For oscillatory systems, the $\lambda_i$ are complex. The pole locations govern the system response. Poles inside the unit circle, $|\lambda_i| < 1$, give stable and convergent responses and are also called damped modes. Poles outside the unit circle, $|\lambda_i| > 1$ have



unstable responses. When a pole is on the unit circle, $|\lambda_i| = 1$, has a sustained oscillation (lossless), referred to as undamped.
In this case, the state variable $x_i$ oscillates sinusoidally at some frequency $\omega_i$, where $\lambda_i = e^{j\omega_i T}$.

The observed parts of the ith system eigenvector $\{\phi_i\}$ is the mode shape $\{\Psi_i\}$ at the sensor locations given by:

$$\{\Psi_i\} = [\bar{C}_M]\{\Phi_i\}. \tag{8}$$

Both the state and the output measurements can be reconstructed using the so-called one-step-ahead predictor using the identified model $(\bar{A}_M, \bar{C}_M)$ and the Kalman filter $(\bar{K}_M)$.

$$\hat{x}_{k+1} = (\bar{A}_M - \bar{K}_M \bar{C}_M)\hat{x}_k + \bar{K}_M y_k, \tag{9}$$

$$\hat{y}_k = \bar{C}_M \hat{x}_k. \tag{10}$$

As a means of cross-validation, different datasets were used for identification and for validation. As a quality measure, we have used the variance-accounted-for (VAF), which gives a measure of how well the linear model predicts the variability of the
output signal and is expressed as:

$$VAF_k = \left(1 - \frac{\text{Var}(y_k - \hat{y}_k)}{\text{Var}(y_k)}\right) \times 100\%, \tag{11}$$

where $\hat{y}_k$ is the output predicted by the identified model for the $k^{th}$ sensor, $y_k$ is the actual measurement for the $k^{th}$ sensor, and Var denotes the variance.

## 3 EXPERIMENTAL SETUP

This section describes the experimental setup used for the present study. First, the main characteristics of the gearbox used for identification are described. Then, details of the fiber-optic strain sensors used and their location on the outer surface of the ring gears are shown. Lastly, the test bench used and the specifications of the tests performed for identification and validation are presented.

### 3.1 Gearbox description

The wind turbine gearbox used for the present study is a four-stage gearbox manufactured by Gamesa Gearbox with a reference torque of 8 MNm. It is considered a suitable example of the gearbox architecture expected to dominate the high-end power ratings, see Sec. 1. The gearbox has a configuration comprised of three planetary stages followed by a parallel helical gear stage, which provides a total gear ratio of 179.576. Figure 1 shows the arrangement of all the stages in the gearbox with the rotor on the left side of the picture. For clarity, only the first stage ring gear has been fully drawn. The first input stage is a
planetary stage with seven planets and has a ring gear with an outer diameter of 2107 mm. The first stage sun is connected





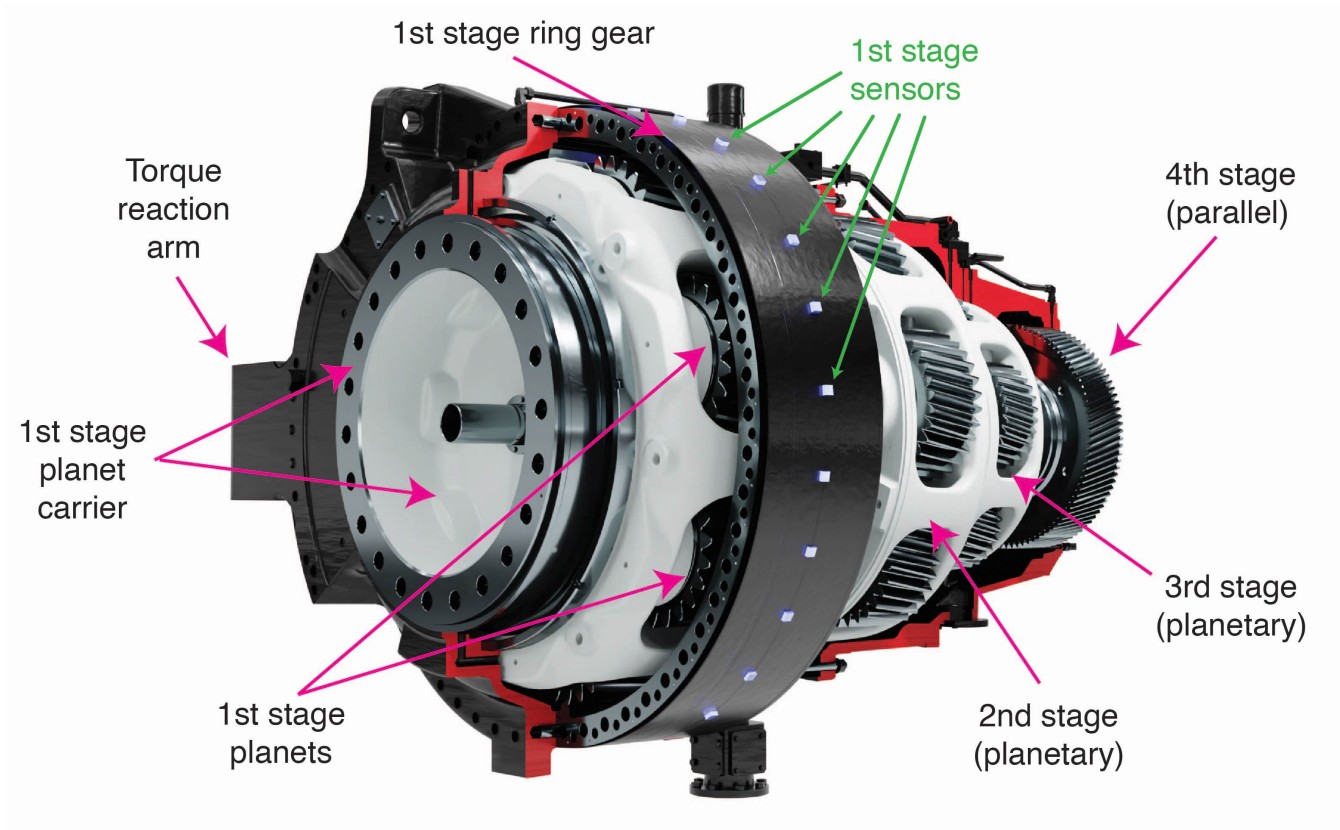

**Figure 1.** 3D Representation of the tested gearbox with fiber-optic strain sensors on the outer surface of the first stage ring gear. Adaptation of figure from Gamesa Gearbox (https://www.gamesagearbox.com/wind-technology/).

with a spline to the second stage planet carrier that contains six planets. The outer diameter of the second stage ring gear is 1790 mm. The third stage has five planets and a ring gear with an outer diameter of 1428 mm. The total weight of the gearbox is approximately 38950 kg, which yields a torque density of 205 Nm/kg. The first and second-stage planets are supported by journal bearings instead of roller element bearings due to the space constraints created by the very large number of planets. The

known excitation frequencies can be computed using the rotational speed and the number of teeth of the gears. The rotational frequencies of the planet carriers, the planet passing frequencies of each stage, and the gear mesh frequencies have been summarized in Table 2. These frequencies correspond to the nominal speed of 8.35 rpm in the first stage planet carrier, which is the low-speed or input shaft.

### 3.2 Fiber-optic strain sensors

Three arrays of fiber-optic strain sensors based on fiber Bragg gratings (FBGs) were wrapped around the planetary stage ring gears. The sensor placement was designed taking into account the insights gained in Gutierrez Santiago et al. (2022), which



**Table 2.** Rotational and gear mesh frequencies of the gearbox at nominal speed.

| Acronym | Description | Frequency (Hz) | Order of LSS |
|---------|-------------|----------------|--------------|
| PC1 | Rotational frequency of first planet carrier | 0.1392 | 1 |
| 7xPC1 | Planet passing frequency of first stage | 0.9748 | 7 |
| PC2 | Rotational frequency of second planet carrier | 0.4603 | 3.3056 |
| 6xPC2 | Planet passing frequency of second stage | 2.7619 | 19.8334 |
| PC3 | Rotational frequency of third planet carrier | 1.7169 | 12.3288 |
| 5xPC3 | Planet passing frequency of third stage | 8.5843 | 61.6441 |
| HSIS | Rotational frequency of high-speed gear wheel | 7.5542 | 54.2468 |
| HSS | Rotational frequency of high-speed gear pinion | 25.0070 | 179.5758 |
| GMF1 | First stage gear mesh frequency | 11.5583 | 83 |
| GMF2 | Second stage gear mesh frequency | 46.4922 | 333.8614 |
| GMF3 | Third stage gear mesh frequency | 145.9335 | 1047.9504 |
| GMF4 | Fourth stage gear mesh frequency | 725.2035 | 5207.6966 |

demonstrated that because the rims are relatively thin significant strains can be measured on the outer surface of the ring gears. In total, 12 optical fibers were installed on the test gearbox, four on each ring gear. A number of grooves were machined on the external diameter of the ring gears, in the middle section relative width of the gear between the rotor and generator side faces, to facilitate the installation process and protect the sensors during assembly and testing. Machining the grooves by turning provided a smooth finish that guaranteed an adequate bonding between the fiber and the ring gear. Figure 2 shows the detailed location of the 42 strain sensors distributed on the outer perimeter of the first stage ring gear. The number of sensors was defined as a multiple of the planets, equally spaced around the perimeter, to ensure that the mesh events caused by the seven planets could be detected synchronously by the strain sensors. The labels of the strain sensors have been colour-coded to represent in which fiber the FBG was accommodated. The spacing between FGBs within each fiber was designed so that all fibers cover the complete perimeter of the ring. This was done to prevent losing a portion of the ring gear in case of damage to a fiber. However, all the fibers survived the complete measurement campaign satisfactorily, including assembly and disassembly operations. Sensor placement on the second and third stage ring gears is shown in Figures 3 and 4, respectively. The fiber optical sensors were supplied and installed by the company Sensing360 B.V. (sensing360.com). For a detailed description of the measurement principle and properties of fiber-optic strain sensors based on FBGs, the interested reader is referred to previous work by Gutierrez Santiago et al. (2022).

In each of the planetary stages, in addition to the fiber-optic strain sensors, inductive displacement sensors were installed to provide a pulse once-per-revolution of the planet carrier. The purpose of these sensors was to know the planet carrier's relative position to the strain sensors to identify which planet is responsible for the strain peaks observed in the strain signals. The relative position of the target and the inductive sensor or pick-up are shown in Figures 2 to 4. During the experiments, torque



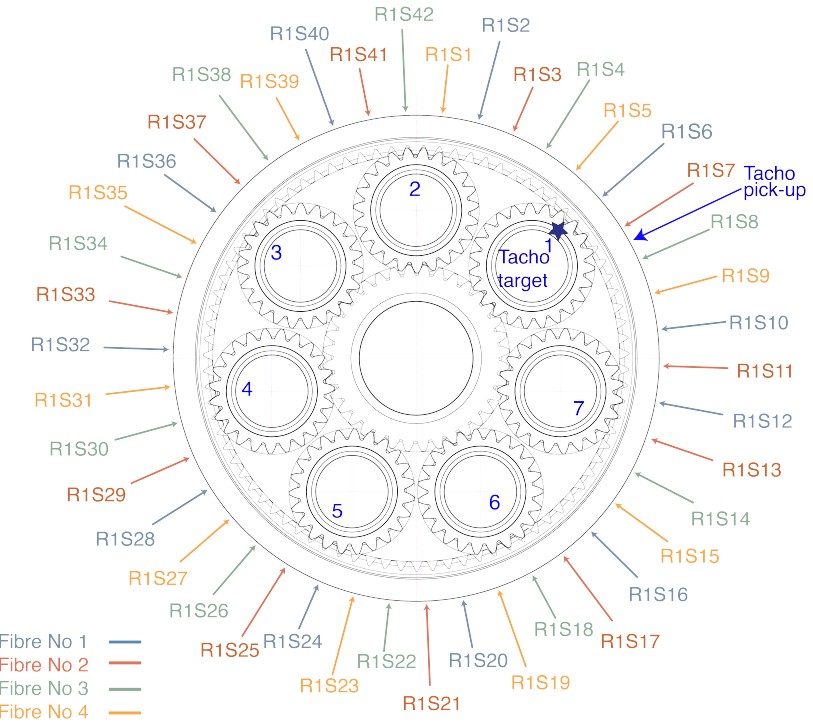

**Figure 2.** Sensor placement on the first stage ring gear.

measurements from torque transducers installed in the high-speed shaft coupling of the test bench were logged synchronously with the fiber-optic strain data and the tachometer signals of all 3 stages.

## 3.3 Test specification

The tests presented in this study were performed on an end-of-line test bench at the assembly factory of Gamesa Gearbox
(Siemens Gamesa Renewable Energy) in Lerma, Spain. The standard IEC 61400-4 (2012) sets the design requirements for wind turbine gearboxes and establishes a mandatory requirement to perform a loaded end-of-line test for all gearbox units before their installation in a wind turbine. This test is also referred to as the run-in or gearbox conditioning test. The purpose is twofold: on the one hand, it serves as a conditioning test for bearings and gears because the gearbox is loaded progressively up to nominal torque; on the other, it provides a means for quality control. Typically, gearbox manufacturers employ a back-
to-back arrangement for end-of-line testing where two gearboxes are connected through the low-speed shaft (LSS). Figure 7 shows the layout of the back-to-back arrangement used for the experiments with the test gearbox, presented in Sec. 3.1, on the left side. An electric motor provides the driving motion to the high-speed shaft (HSS) of one gearbox, and the other motor acts as a generator, providing the barking torque at the HSS of the second gearbox. The rated power of the test bench electric motors is 11.5 MW which enabled testing the gearbox above its nominal torque. Although the test bench is designed to recreate





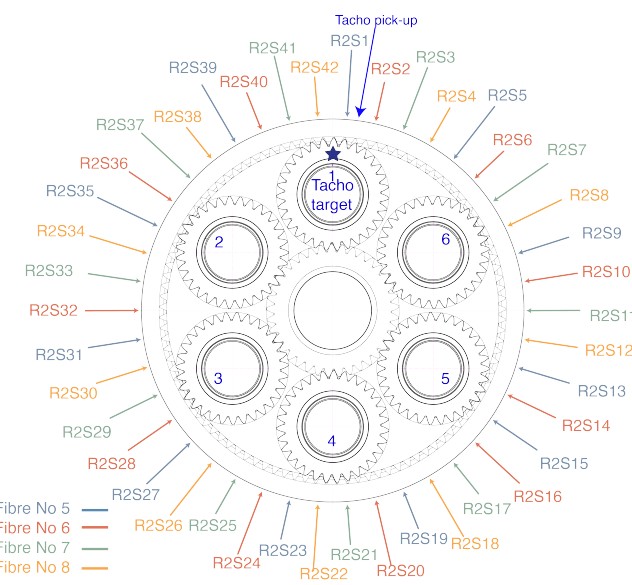

**Figure 3.** Sensor placement on the second stage ring gear.

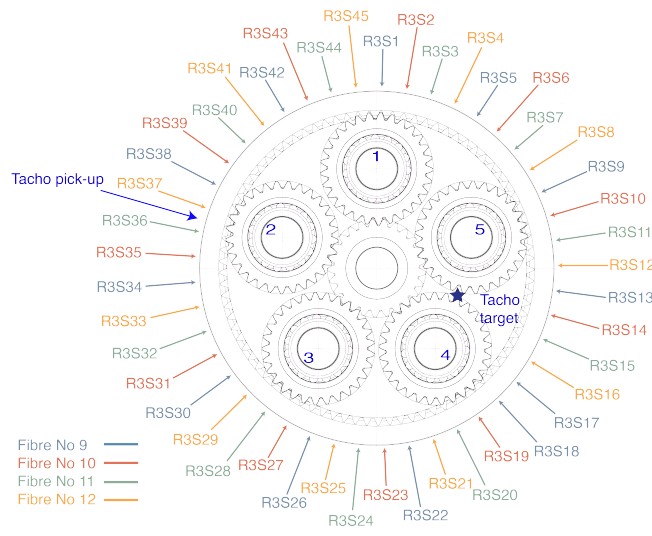

**Figure 4.** Sensor placement on the third stage ring gear.

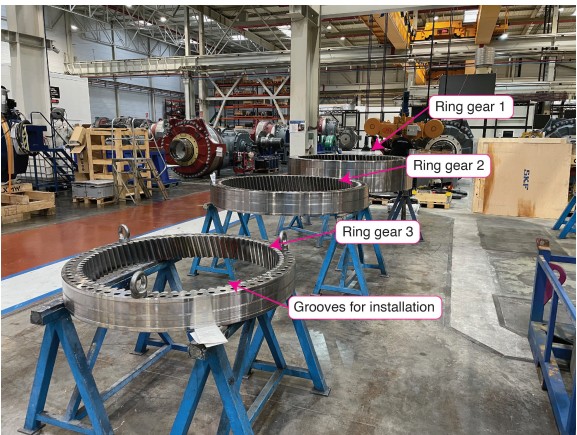

**Figure 5.** All three ring gears with machined grooves ready for sensor installation.

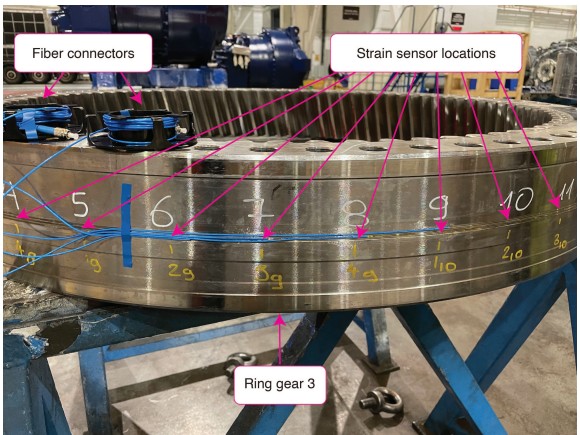

**Figure 6.** Detail of fiber-optic sensors installed on the third stage ring gear.

the working conditions of the gearbox as close as possible to the wind turbine, in back-to-back test benches, torque is the only controlled input load excitation, and it is designed not to apply bending moments to the gearbox. The mechanical interfaces at the LSS and HSS of the gearbox are different from the wind turbine, and it is not possible to reproduce the rotor inertia in the test bench. Despite these differences, we consider the back-to-back test bench results representative of the behavior of



the gearbox in a wind turbine. Specially taking into account that these gearboxes are designed for operation in wind turbine
drivetrains with a four-point mount suspension.

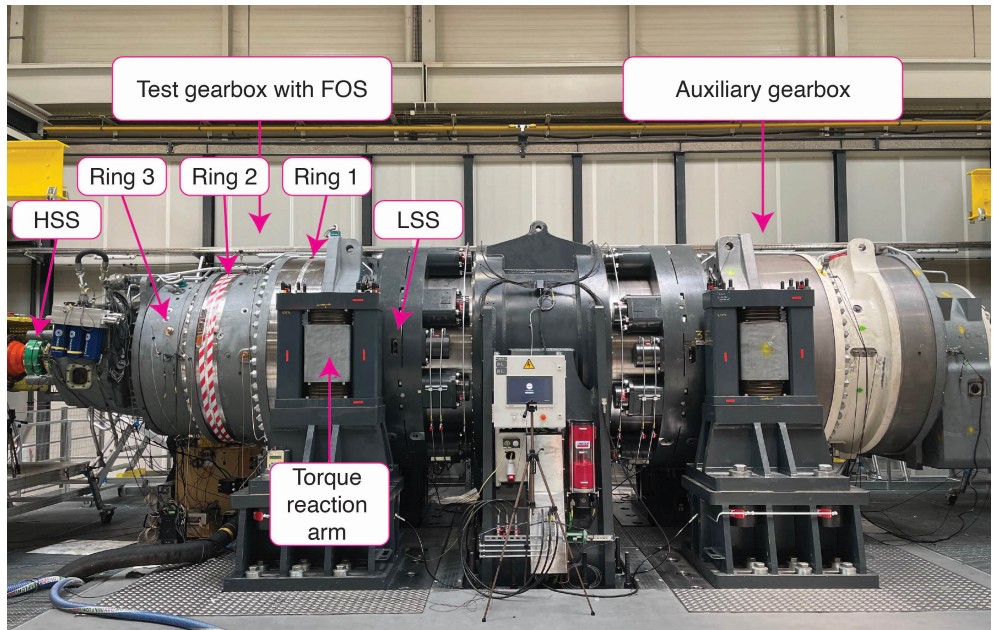

**Figure 7.** Test gearbox on the left side of the back-to-back test bench used for end-of-line testing of wind turbine gearboxes with a maximum power capacity of 11.5 MW.

The instrumented gearbox completed a standard end-of-line test, composed of six stationary load stages under nominal speed. Once stable thermal conditions had been reached, signals from the fiber-optic strain sensors were logged at each of the run-in load stages to perform system identification. After the run-in, several design validation tests were performed, and these tests were used to collect more strain data to evaluate the state and output estimation procedures. In particular, a test to validate
the structural models of the gearbox, comprised of 22 stationary torque conditions from 5% to 110% of its nominal value, was used to evaluate the effect of torque on the identified operational deflection shapes. Finally, different tests with dynamically changing torque were performed to quantify the contribution of the identified deflection shapes in a dynamic manner.

## 4   Identification of operational deflection shapes

This section describes the key findings obtained when performing system identification on the strain signals logged during
experiments performed on a serial production end-of-line test bench.



## 4.1 Identification using signals from all stages

The system identification framework presented in Section 2 was initailly applied to all available signals from the three ring gears together. Figure 8 shows a time trace of two strain sensors from each stage from a test performed with rated stationary speed and torque conditions. As can be seen, each stage has a different rotational speed and the deformations caused by the
mesh forces between the planets and the ring gears occur at different times. The time interval between strain peaks corresponds to the planet passing frequencies defined in Table 2. For clarity, only two sensors from each ring gear have been plotted in Fig. 8 but there are 42 strain signals available in the first stage, 42 in the second stage and 45 in the third stage. The location of all sensor are shown in Figures 2, 3 and 4.

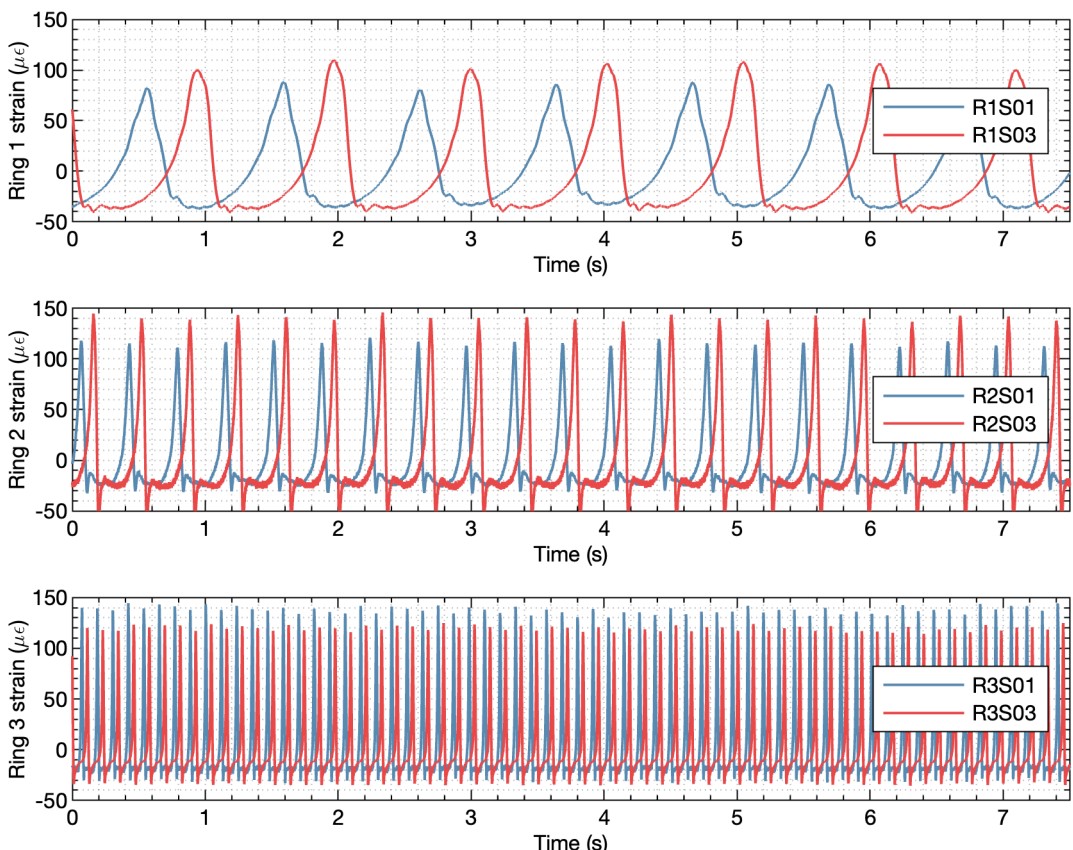

**Figure 8.** Raw strain signals logged from the first and third sensors of each planetary stage. The angular location of the sensor R1S01, R1S03, R2S01, R2S03, R3S01 and R3S03 is shown in Figures 2, 3 and 4.





Several data preprocessing steps were performed on the raw signals logged by the optical interrogators. First, the sampling
frequency was down-sampled for a more efficient numerical implementation. Generally, the sample rate should be up to about
ten times the bandwidth of interest to avoid the effects of aliasing and, simultaneously, limit the amount of high-frequency noise
that contaminates the measurements (Verhaegen and Verdult, 2007). The optical interrogators used to acquire the signals during
testing provided a sampling frequency of 2000 Hz in the first stage and 2500 Hz in the second and third stages. Considering the
known excitation frequencies present during gearbox operation, see Table 2, different downsampled frequencies from 45 Hz
to 250 Hz were tested. The difference between resampling, interpolation, and decimation on the identified parameters was
found to be negligible. Therefore, resampling with an embedded antialiasing filter was chosen as the downsampling method.
Measurements used for identification were logged once the gearbox had reached thermal stability. Thus, the influence of
temperature variations on the FOS signals was minimized. Nevertheless, a detrending step was added to ensure that the signals
fed to the identification algorithm only resulted from the strain caused by the gear mesh events. All signals were normalized
using their standard deviation to have unit variance. As a last preprocessing step, a hamming window was applied to the training
data sequences because it was found to reduce the variance of the identified models.

As described in Sec. 2, from sequences of discrete-time data samples of the measured signals, three parameters need to
be defined to execute the MOESP algorithm. These parameters are the number of samples $N$, the number of block rows $s$,
and the system order $n$. Using fiber-optic strain signals from the total of 129 sensors (42 from the first stage, 42 from the
second and 45 from the third) different options for $N$, $s$, and $n$ were explored. The integer $s$ should be chosen to be about 2-3
times the maximum expected model order (Verhaegen and Verdult, 2007). The experiment duration, number of samples $N$,
should usually be at least about ten times the length of the slowest time constant of the system to ensure that the low-frequency
behavior of the process is captured. Therefore, a trade-off between sample frequency and measurement duration must be made
that is dictated by storage and/or processing limitations regarding the number of data points. After exploring different down-
sampled frequencies between 45 and 250 Hz, a frequency of 62.5Hz was selected. This selection was based on identified
frequencies and the signal reconstructions obtained using the one-step ahead predictor Eq. 10. Figure 9 shows the discrete-
time representation of the pole locations of the identified models using $N = 17500$ samples per signal, $s = 64$ block rows and
$n = 20$ a model order equivalent to 10 oscillatory modes. All identified poles are on the unit circle, which is expected from
the periodic behavior. The corresponding frequencies associated with the identified poles are shown in Table 3. All identified
frequencies match with known excitation frequencies. A description of the acronyms used to name the frequencies can be found
in Table 2. The term operational deflection shapes (ODS) has been chosen for the observed part of the identified eigenvectors
because they are caused by periodic excitations and not a structural property of the gearbox. These deflection shapes identified
when using all strain signals together only influence one ring gear at a time. To illustrate this, the three mode shapes related
to the planet passing frequencies of each stage are shown in Figure 10. For example, in the case of the mode associated with
the planet passing of the first stage, with an identified frequency of 0.9750 Hz, the deformations of this mode shape in the
second and third stages are negligible. This means there is very little cross-stage excitation, which is positive and one of the
design objectives. Considering these results, it was decided to apply the identification algorithm on strain data from each stage
individually.





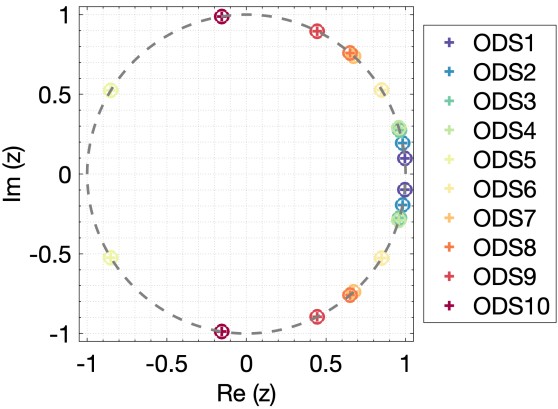

**Figure 9.** Pole locations (eigenvalues) of the identified model using measurement signals from all three stages, discrete-time representation ($N = 17500$, $s = 64$, $n = 20$).

**Table 3.** Identified frequencies using signals from all three stages, description of acronyms can be found in Table 2.

| Mode | Frequency (Hz) | Order of LSS | Acronym |
|------|----------------|--------------|---------|
| 1 | 0.9748 | 7.0000 | 7xPC1 |
| 2 | 1.9496 | 13.9999 | 2x7xPC1 |
| 3 | 2.7618 | 19.8327 | 6xPC2 |
| 4 | 2.9244 | 21.0000 | 3x7xPC1 |
| 5 | 5.5234 | 39.6634 | 2x6xPC2 |
| 6 | 8.2861 | 59.5023 | 3x6xPC2 |
| 7 | 8.5844 | 61.6445 | 5xPC3 |
| 8 | 11.0466 | 79.3258 | 4x6xPC2 |
| 9 | 17.1688 | 123.2890 | 2x5xPC3 |
| 10 | 25.7532 | 184.9335 | 3x5xPC3 |

## 4.2 Identification using signals from the first planetary stage

Using data from the same test, with rated torque and speed conditions shown in Figure 8, the system identification procedure was repeated with the strain signals from the first stage ring gear only. The same preprocessing steps detailed in Section 4.1 were applied. Ten different down-sampled frequencies (45.45, 50.00, 55.55, 62.50, 71.43, 83.33, 100.00, 125.00, 166.67, and 250.00 Hz) were tested to explore the effect of resampling on the identified models using a baseline setting of $s = 32$ block rows and $n = 20$ model order. The number of samples was chosen to cover the same training time, defined as 256 s, in all

sampling frequencies. The different identified models were evaluated based on their identified frequencies (eigenvalues) and how well newly measured data from validation tests could be reconstructed using the identified operational deflection shapes. As in the case of all stages, the difference between identified frequencies using different sampling frequencies was found to be small. Again, all identified poles were on top of the unit circle, corresponding to undamped modes. To evaluate the accuracy of the reconstructed outputs, data for validation was acquired using the same experimental conditions as for training

the models: applying stationary rated torque and speed and waiting for temperatures across the gearbox to stabilize. With the identified systems matrices, the system's initial state can be estimated, and the state and output signals can be reconstructed using Eq. 10. For model validation, the Kalman filter was not considered. In this case, the system's behavior was modeled as an autonomous system oscillating from a non-zero initial condition. Figure 12 shows a comparison of the measured signal from sensor R1S01 (first sensor of first stage ring gear) against the reconstructed output using identified models with three

different sampling frequencies. Very high variance accounted for (VAF) values were obtained with the reconstructed outputs.

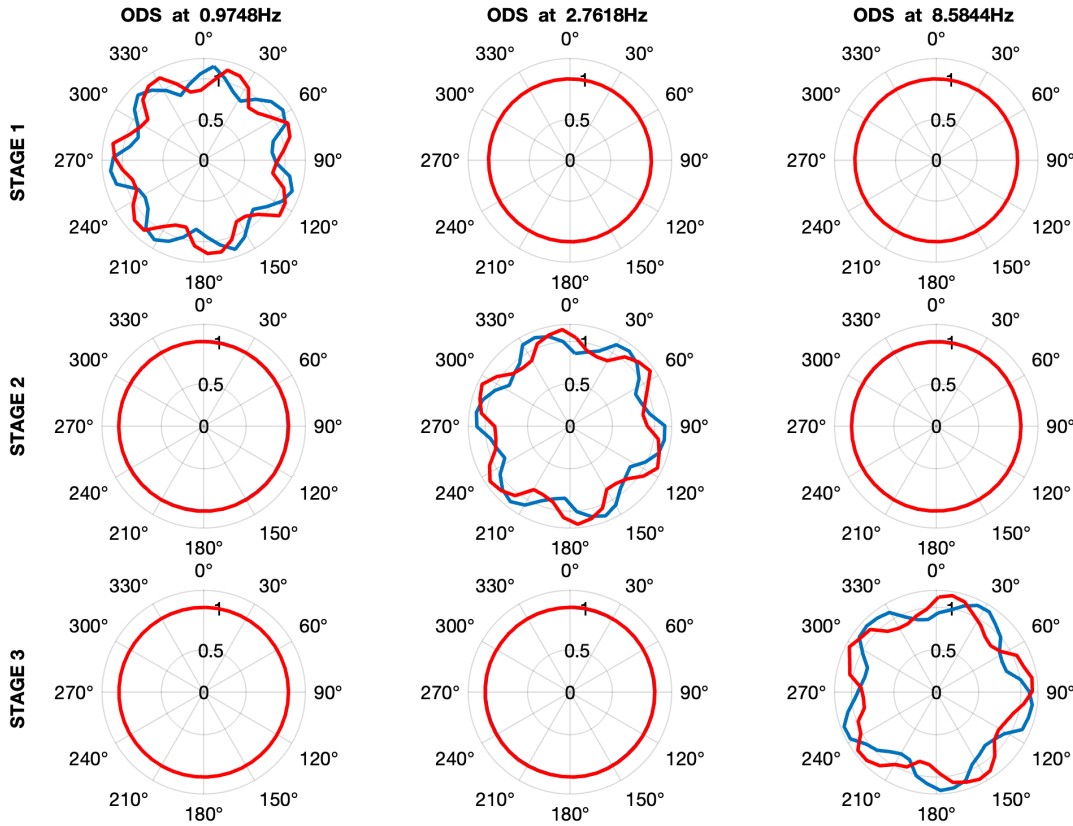

**Figure 10.** Identified operation deflection shapes using measurement signals from all three stages.

The highest average VAF was 99.68% obtained for a sampling frequency of 62.5 Hz and the lowest value was 98.40% for the case of 250 Hz. For this reason, 62.5 Hz was used as a sampling frequency to search for suitable $s$ and $n$ parameters. Table 4 shows the average VAFs obtained for different combinations of $s$ block rows and $n$ model orders. The VAF value presented is the average of the 42 sensors. Increasing the model order and the number of block rows improved the fit between the

reconstructed and measured signals, an average VAF value of 99.00% was already achieved with $s = 32$ and $n = 10$. This VAF could be increased up to 99.90% when further increasing $s$ and $n$ to $s = 128$ and $n = 32$. As with other practical applications of system identification (Hermans and van der Auweraer, 1999), we did not observe a big gap in VAFs from one model order to the next. Table 5 shows the frequencies of the 10 deflection shapes identified using $s = 128$ and $n = 20$. Up to $n = 18$, all the deflection shapes correspond to multiples of the planet passing frequency (seven times the carrier rotational frequency), and for

larger model orders, the planet carrier rotational frequency is also identified. Figure 11 shows the deflection shapes associated with the planet passing frequency and the first two integer multiples or harmonics. Due to the given spatial resolution (42 sensors around the ring gear), it is not possible to represent higher frequency mode shapes accurately as they provoke a spatial





aliasing effect. This, however, does not affect the output reconstruction of individual sensors as long as the sampling frequency is high enough for the identified modes.

**Table 4.** VAF of stage 1 strain measurements and reconstructions with a sampling frequency of 62.5 Hz and 16000 samples.

| VAF (%) | $n=2$ | $n=6$ | $n=10$ | $n=14$ | $n=18$ | $n=20$ | $n=24$ | $n=32$ |
|---|---|---|---|---|---|---|---|---|
| $s=8$ | 3.04 | 26.67 | - | - | - | - | - | - |
| $s=16$ | 60.02 | 69.66 | 98.83 | 98.99 | - | - | - | - |
| $s=32$ | 72.42 | 97.49 | 99.00 | 99.39 | 99.34 | 99.69 | 99.71 | - |
| $s=48$ | 72.40 | 89.02 | 99.08 | 99.32 | 99.44 | 99.64 | 99.65 | 99.73 |
| $s=64$ | 72.55 | 97.50 | 99.33 | 99.44 | 99.62 | 99.69 | 99.73 | 99.75 |
| $s=96$ | 72.55 | 97.50 | 99.33 | 99.45 | 99.53 | 99.71 | 99.76 | 99.77 |
| $s=128$ | 72.55 | 97.50 | 99.34 | 99.43 | 99.59 | 99.78 | 99.83 | 99.90 |

**Table 5.** Identified frequencies measurement signals from 1st stage using $s=128$ and $n=20$ (10 modes).

| | Frequency (Hz) | Order of LSS | Description |
|---|---|---|---|
| 1 | 0.1407 | 1.0108 | PC1 |
| 2 | 0.9748 | 7.0000 | 7xPC1 |
| 3 | 1.9496 | 14.0000 | 2x7xPC1 |
| 4 | 2.9244 | 20.9999 | 3x7xPC1 |
| 5 | 3.8992 | 27.9999 | 4x7xPC1 |
| 6 | 4.8739 | 34.9993 | 5x7xPC1 |
| 7 | 5.8487 | 41.9997 | 6x7xPC1 |
| 8 | 6.8233 | 48.9984 | 7x7xPC1 |
| 9 | 7.7983 | 55.9998 | 8x7xPC1 |
| 10 | 8.7731 | 62.9998 | 9x7xPC1 |

The reconstructed output signals shown in Figures 12 were computed using the system matrices only, finding the initial state conditions and assuming the system behaves like an autonomous system. We can improve the state estimation using Eq.10 with the Kalman filter. This allows the analysis of strain measurements from tests with variable torque. The states associated with each mode shape convey the contribution of each mode to the measured strain signals. For the validation test performed using stationary rated torque and speed, Figure 13 shows the evolution of the states associated with the operational deflection

shapes described in Table 5. The system matrix was transformed into diagonal form, with the eigenvalues in the diagonal. As described in Sec. 2 these eigenvalues are complex numbers and for oscillatory systems come in conjugate pairs. Therefore, two states, which are also conjugate imaginary numbers, are associated with pair of eigenvalues. The magnitude shown in Figure 13 is the modulus or absolute value of the state variables. From Figure 13 we can infer that the contribution of the first deflection



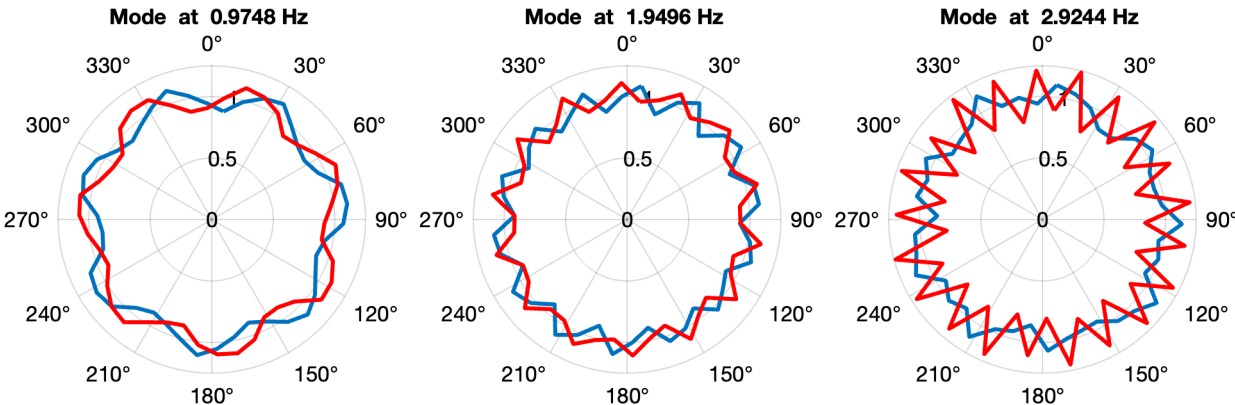

**Figure 11.** Mode shapes from first three harmonics of planet passing frequency using measurement signals from the first stage.

shape, related to the rotation of the planet carrier, is relatively small. The second deflection shape, created by the passing of the
planets at seven times planet carrier frequency, is the most dominant mode shape, and its higher harmonics have a descending
contribution.

## 4.3 Effect of torque on identified models and state variables

Once a model has been identified using suitable training data, that is, the operational deflection shapes and their frequencies
have been found, the associated states can be computed using the one-step-ahead predictor, Eq. 9. The states presented in
Figure 13 correspond to a test with rated torque and speed where the identified ODS, the measured signals, and the Kalman
filter were used to reconstruct the states. These state variables remain almost constant and exhibit small changes. These small
changes are also evident in the test bench torque signals and in the peak values of the fiber-optic strain signals, see Figure 8.
In Figure 14, an animation of the strains reconstructed using only the first three operational deflection shapes associated with
the passing of the planets is shown. To evaluate the effect of torque on the identified deflection shapes, a test comprised of
22 stationary torque conditions from 5% to 110% of its nominal value was performed. This test was originally intended to
validate the structural models of the gearbox. Once stability in torque and speed was reached, data was recorded for 240 s for
each torque condition. Torque data from two test bench torque sensors installed in the high-speed shafts (HSS) was logged
synchronously with the fiber-optic strain data. From these two sensors, the torque at the low-speed shaft (LSS) was estimated
as the average value of both high-speed shafts multiplied by the gear ratio. This assumes that the gear losses are equal in both

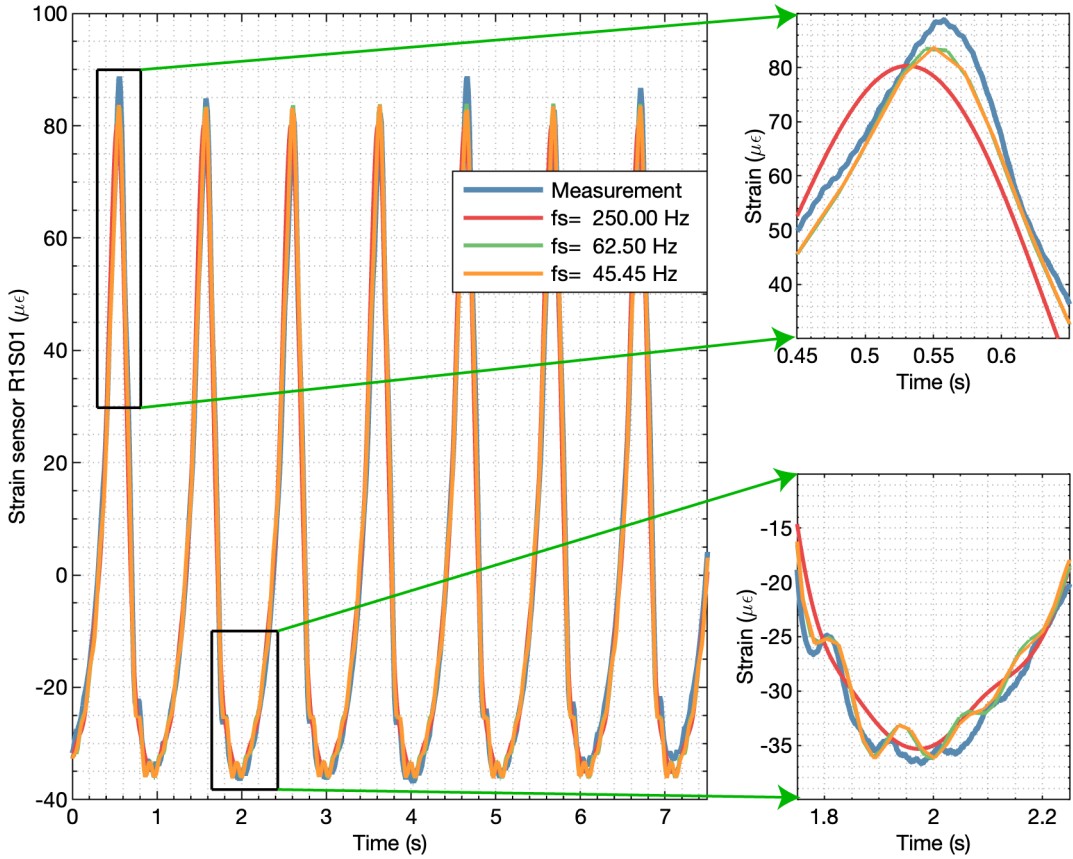

**Figure 12.** Output reconstruction of strain signal R1S01 using models identified with different sampling frequencies.

gearboxes, which is not exactly true because the two gearboxes tested were not identical, and the torque level in the gearbox acting as a reducer is slightly higher, but it is considered a good approximation to evaluate the effect of torque.

The 22 data recordings at different torques were used to identify operational deflection shapes. Figure 15 shows the deflection shapes of the mode corresponding to the planet passing frequency of the first stage (seven times the rotational frequency of the carrier) from 55% to 100% of the nominal torque. When the system matrix is transformed into diagonal form, as described in Sec. 2, each mode $\{\phi_i\}$ comes in conjugate pairs of imaginary numbers. For clarity, only the real component of the mode shape is shown in a linear format, and the magnitudes have been normalized using the norm of the deflection shape at nominal torque. The shapes are very similar, with only very slight differences observed when the torque drops below 65% of nominal torque. The gearbox is designed to operate in near-rated torque conditions where the gear microgeometry has been optimized.

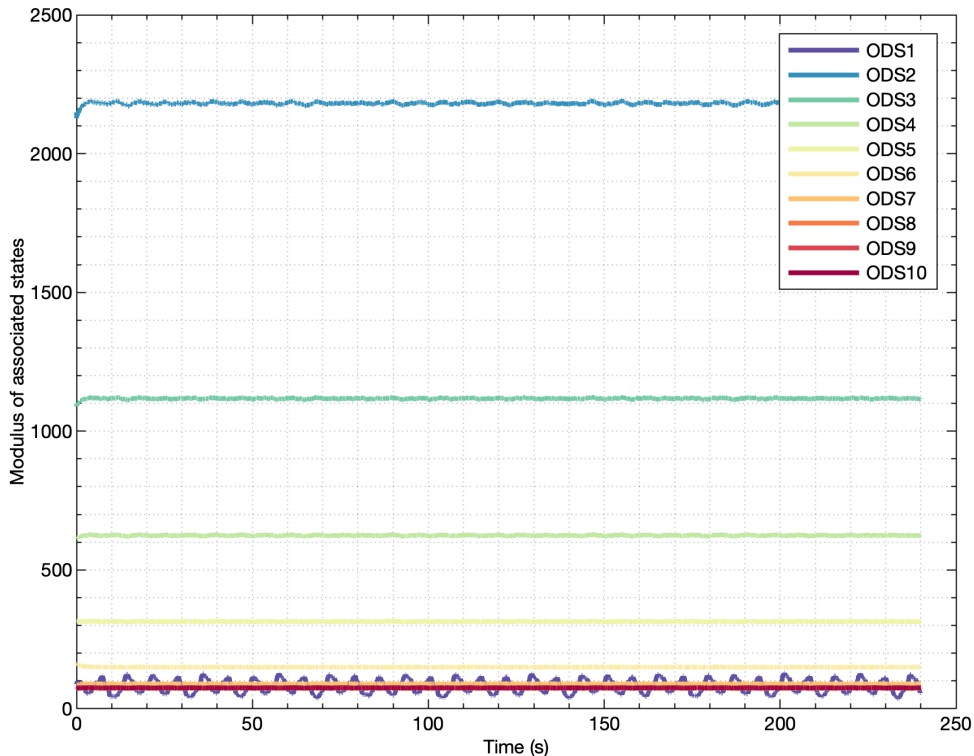

**Figure 13.** Module of states associated with each ODS during the structural model validation tests at nominal speed and torque.

This observation led to evaluating the relationship between the contribution of mode shapes identified at nominal torque
conditions for the 22 load stages. Using the identified ODS at nominal torque, the corresponding states were computed for the
data recordings at different toques. When using the diagonal form, the state variables are also conjugate imaginary numbers.
The modulus or absolute value of the state variables for each test against torque is shown Figure 16. The modulus or absolute
value of the states associated with the first deflection shape doesn't exhibit any relationship with torque. However, all the states
associated with the planet passing frequency and its harmonics show a very strong relationship with torque. A polynomial fit
was computed between the module of the state and torque, which can be used to estimate torque from a known state value.
To demonstrate this we performed a test with 6 torque levels. In Figure 17, the torque estimation from the test bench torque
sensors is compared to the torque estimation using the state variable associated with the planet passing of the first stage. As
can be seen, the torque estimate using the planet passing mode closely follows the behavior of the torque estimate from the test
bench torque sensors with a similar pattern. As mentioned before, the torque sensors are placed in the HSS of both gearboxes
in the back-to-back arrangement, and a comparison with a direct measurement in the input LSS is suggested to further evaluate
the accuracy of the new estimation method.

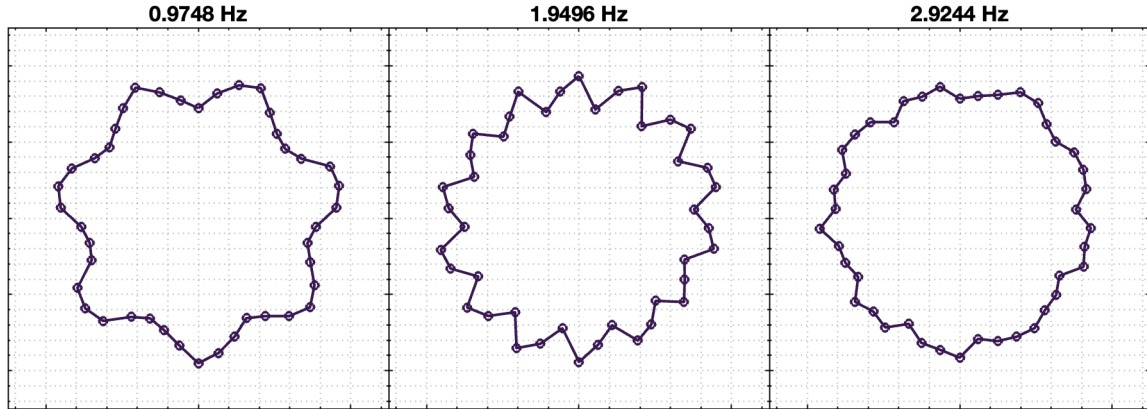

**Figure 14.** Simulated strains on the first stage ring gear associated with the first three planet passing modes, animation of a planet carrier revolution can be accessed in supplementary video file.

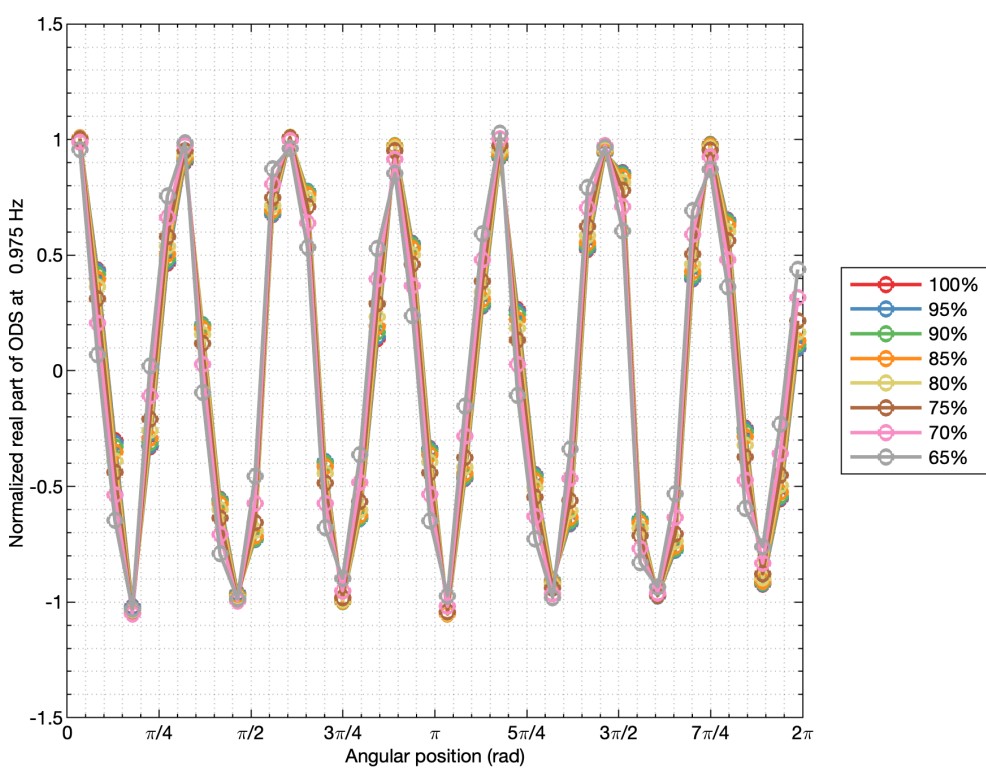

**Figure 15.** Operational deflection shapes (real part) using datasets at different torque levels for identification.

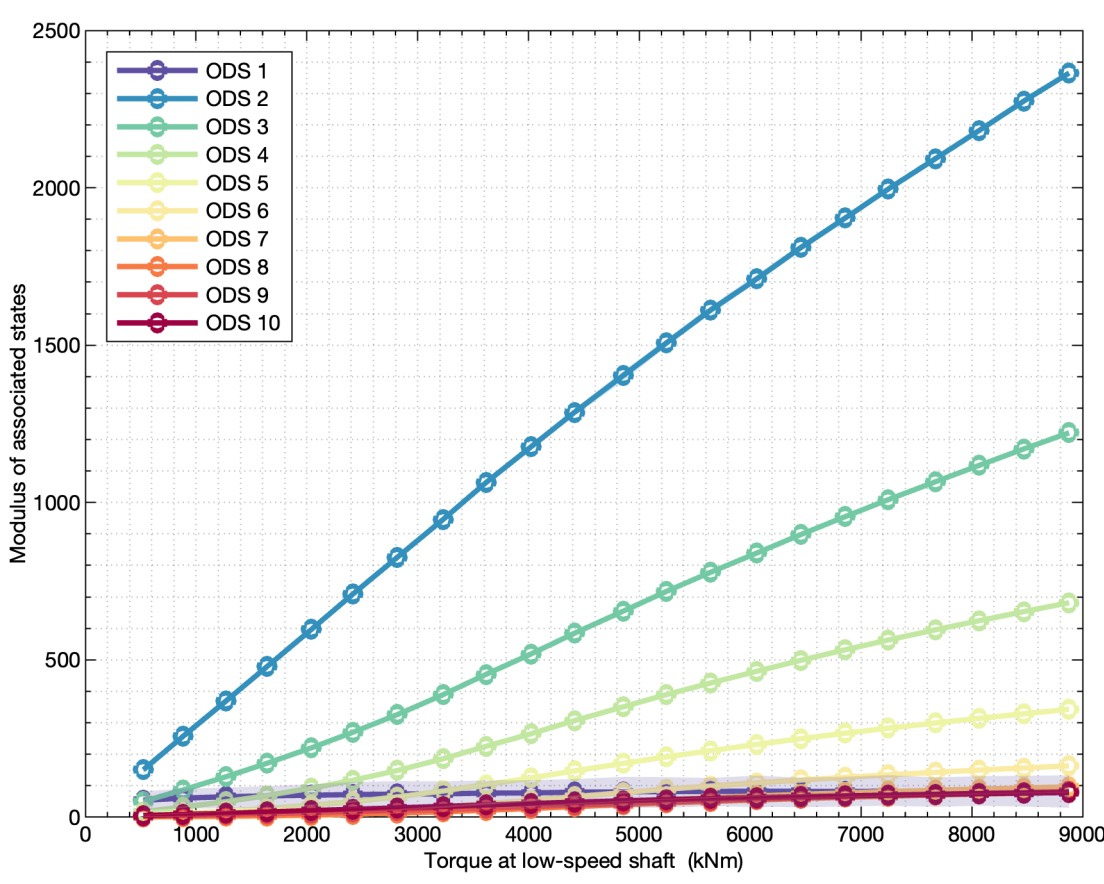

**Figure 16.** Average modulus of states associated with each ODS against low-speed shaft torque (from test bench sensors).

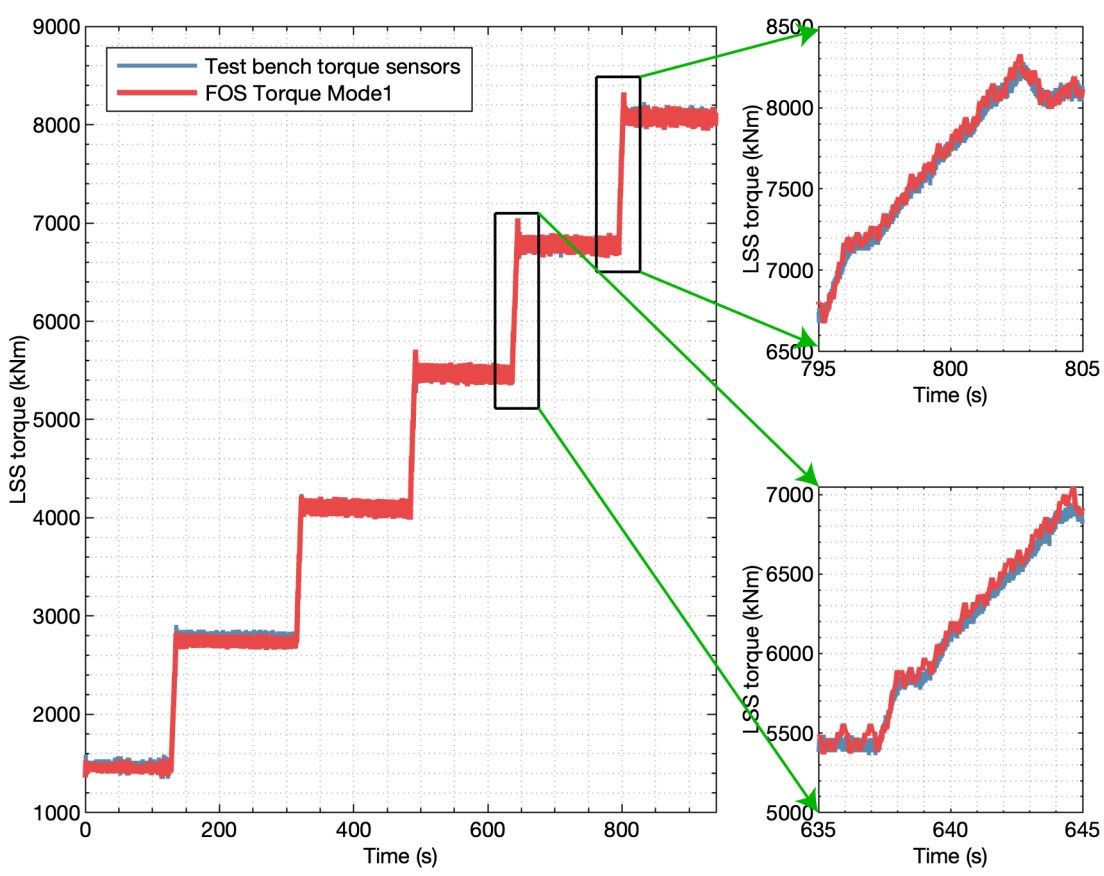

**Figure 17.** Torque estimation from identified ODS vs test bench torque sensors for a test with dynamic torque conditions.





# 5 Conclusions

This article presents a novel measurement setup of 129 fiber-optic strain sensors distributed around the three ring gears of a modern wind turbine gearbox. The subspace identification multivariable output-error state space method (MOESP) has been

applied to experiments performed on a serial production end-of-line test bench and found to provide consistent estimates. Using signals from tests with stationary torque and speed conditions, all identified eigenvalues and eigenvectors correspond to periodic excitations related to shaft rotations, planet passing, and gear mesh frequencies. When performing system identification on data from all three stages, the identified deflection shapes have been found to cover only one stage at a time. Therefore, no cross-excitation between stages was observed, which is in line with one of the design intents to minimize cross-stage interactions.

Therefore, it can be concluded that the identification algorithm can be applied to strain data from each stage individually.

For each planetary stage, the effect of the different identification parameters that can be chosen in the MOESP algorithm has been explored. Measurements from model validation tests, with the same torque and speed conditions as the ones used for identification or training, have been used to evaluate the identified frequencies and mode shapes. The variance accounted for (VAF) between the validation measurements and the reconstructed outputs, simulating the system's behavior as an autonomous

system oscillating from a non-zero initial condition, has been used as a metric. For the signals of the first stage ring gear, average VAF values above 99% were achieved between the signals measured in the validation tests and the reconstructed signals for suitable combinations of $s$ block rows and $n$ model order. Therefore, the identified deflection shapes can reproduce the behavior of the gearbox accurately, and the contribution of the periodic excitations accounts for almost all the energy in the measured strain signals. The effect of torque on the identified deflection shapes has been studied, and no noticeable differences

in the planet passing mode shapes were observed for torques above 65% of the nominal value. For strain recordings from tests with dynamically changing torque conditions, the contribution of the periodic modes has been quantified through the states associated with the operational deflection shapes identified at nominal torque. The contribution of the deflection shapes produced by the passing of planets is controlled by the amount of input torque applied to the gearbox. Using this contribution an estimate of the input torque has been demonstrated for dynamic operating conditions.

Accurate knowledge of the input torque is critical to ensuring the reliability of wind turbine gearboxes. It can also improve health monitoring because knowing the real-life torque of wind turbine gearboxes throughout their service life enables a better assessment of the remaining useful life. The system identification framework presented in this article can be applied recursively to track the operational deflection shapes over time, which could lead to early fault detection in the planetary stage components. Three avenues are suggested for future work. First, it is recommended to quantify the accuracy of the torque estimate produced

by the strain measurements in the outer surface of the ring gear against a conventional direct measurement in the input shaft. When assessing the accuracy, the effects of non-torque loads, i.e., axial forces and bending moments, should be explored. Second, we suggest researching different sensor configurations and loading conditions that can excite the structural modes. Finally, it is suggested to investigate the fault detection capabilities of trending the operational deflection shapes. Ideally by seeding known faults in components of the planetary stages and evaluating their impact on the identified mode shapes.



*Code and data availability.* Due to confidentiality agreements with research collaborators, the raw data and the software code used to produce the results shown in this publication can only be made available to researchers subject to a non-disclosure agreement. Details of the data and how to request access are available from the 4TU.ResearchData repository at https://doi.org/10.4121/.

## Appendix A

### A1    Identification using signals from the second and third planetary stages

The same identification exercise presented in Sec. 4.2 was performed for the strain signals acquired for the second and third stages. Using the same approach as for the first stage, first, a suitable sampling frequency was selected and then the effect of the identification parameters $s$,$n$ and $N$ were explored. For the second stage, a sampling frequency of 208.33 Hz was found to give satisfactory identification results. Table A1 shows the frequencies associated with the deflection shapes using $s = 128$ and $n = 20$. In this case, all identified frequencies correspond to the planet passing frequency, six times the carrier rotational

frequency, and its harmonics. The first three identified mode shapes of the second stage are shown in Figure A1, and an animation of the reconstructed strain signals using these deflection shapes is shown in Figure A2. In this case, due to the higher frequency and frames-per-second of the animation had to be reduced and could not match the identified frequencies. For the third stage, a sampling frequency of 625 Hz was chosen, and the identified frequencies are shown in Table A2. In this case, using $s = 128$ and $n = 20$, the first nine identified frequencies correspond to the planet passing harmonics (third stage has five

planets). The last identified frequency corresponds to twice the gear mesh frequency of the second stage. The second stage ring gear drives the third stage planet carrier. However, the contribution of this mode is very small. The first three identified mode shapes of the third stage are shown in Figure A3, and an animation of the strains reconstructed using these deflection shapes is shown in Figure A4. Again, the allowable frames-per-second could not match the identified frequencies and the speed of the animation had to be reduced.

*Author contributions.* Unai conducted the tests and performed the data analysis presented in the manuscript. All authors provided important input to this research work through discussions, feedback, and manuscript improvement.

*Competing interests.* Some authors are members of the editorial board of Wind Energy Science. The authors have no other competing interests to declare.

*Acknowledgements.* We would like to sincerely acknowledge the support of Gamesa Gearbox (Siemens Gamesa Renewable Energy) and
TU Delft which made this research possible and the collaboration with Sensing 360 B.V.. The authors would also like to thank our Siemens Gamesa colleagues who helped during the installation of the sensors and the execution of the tests.



**Table A1.** Identified frequencies using measurement signals from the 2nd stage with $s = 128$ and $n = 20$ (10 modes).

|  | Frequency (Hz) | Order of LSS | Description |
|---|---|---|---|
| 1 | 2.7619 | 19.8330 | 6xPC2 |
| 2 | 5.5238 | 39.6661 | 2x6xPC2 |
| 3 | 8.2856 | 59.4992 | 3x6xPC2 |
| 4 | 11.0475 | 79.3322 | 4x6xPC2 |
| 5 | 13.8094 | 99.1653 | 5x6xPC2 |
| 6 | 16.5712 | 118.9983 | 6x6xPC2 |
| 7 | 19.3331 | 138.8314 | 7x6xPC2 |
| 8 | 22.0950 | 158.6643 | 8x6xPC2 |
| 9 | 24.8581 | 178.5060 | 9x6xPC2 |
| 10 | 27.6304 | 198.4140 | 10x6xPC2 |

**Table A2.** Identified frequencies using measurement signals from the 3rd stage with $s = 128$ and $n = 20$ (10 modes)..

|  | Frequency (Hz) | Order of LSS | Description |
|---|---|---|---|
| 1 | 8.5844 | 61.6445 | 5xPC3 |
| 2 | 17.1688 | 123.2892 | 2x5xPC3 |
| 3 | 25.7531 | 184.9337 | 3x5xPC3 |
| 4 | 34.3375 | 246.5780 | 4x5xPC3 |
| 5 | 42.9218 | 308.2222 | 5x5xPC3 |
| 6 | 51.5062 | 369.8665 | 6x5xPC3 |
| 7 | 60.0901 | 431.5077 | 7x5xPC3 |
| 8 | 68.6750 | 493.1560 | 8x5xPC3 |
| 9 | 77.2774 | 554.9303 | 9x5xPC3 |
| 10 | 91.8262 | 659.4054 | 2xGMF2 |

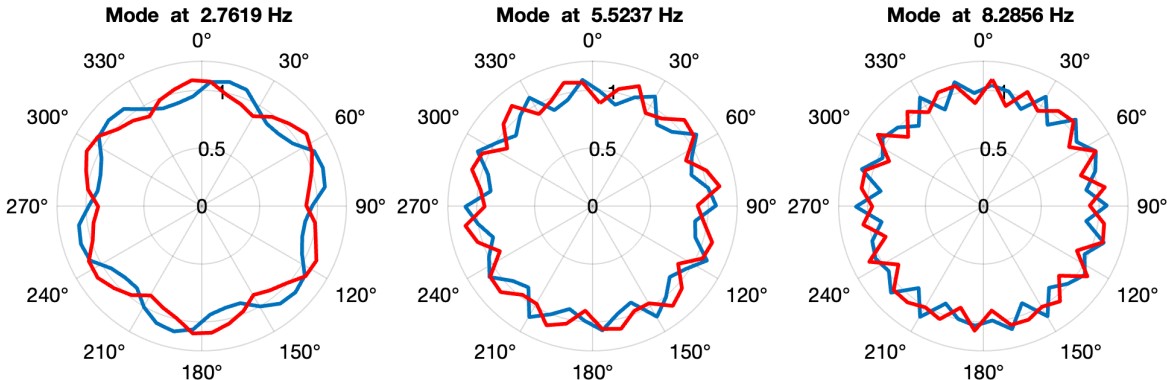

**Figure A1.** Mode shapes from first three harmonics of the planet passing frequency using measurement signals from the second stage.



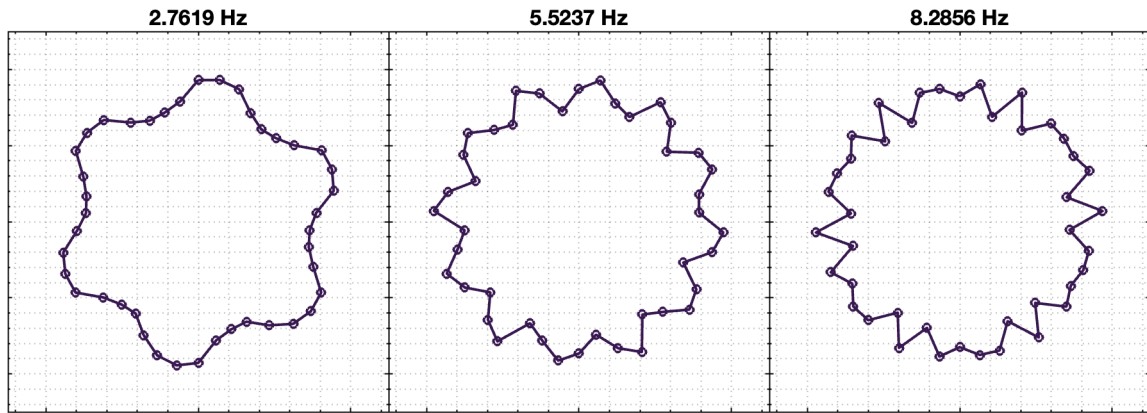

**Figure A2.** Simulated strains on the second stage ring gear associated with the first three planet passing modes, animation of a planet carrier revolution can be accessed in supplementary video file.

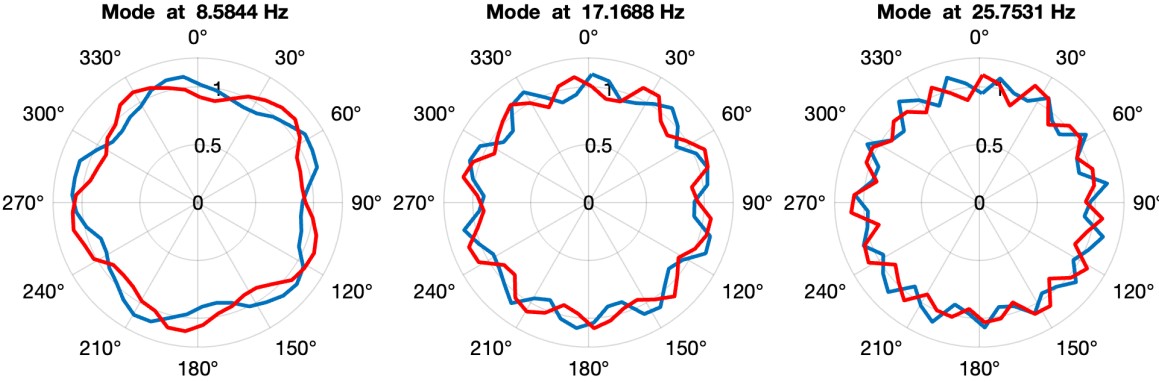

**Figure A3.** Mode shapes from first three harmonics of planet passing frequency using measurement signals from the third stage.

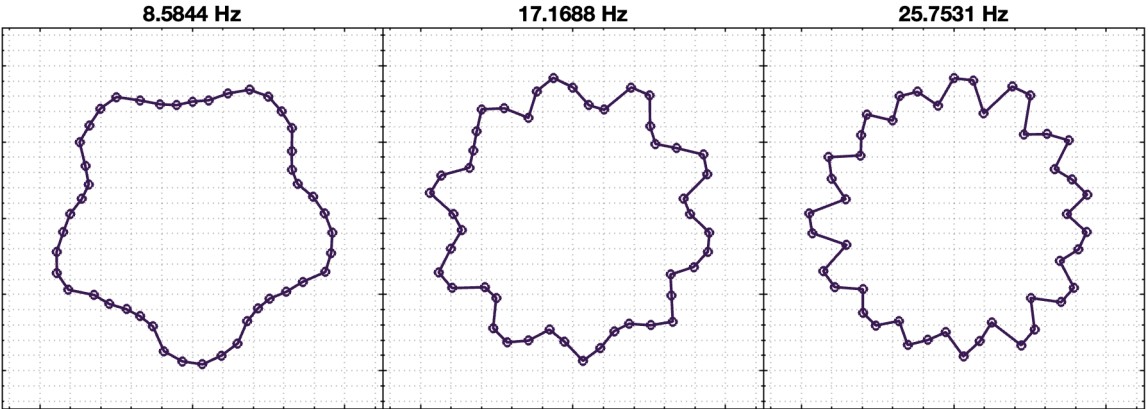

**Figure A4.** Simulated strains on the third stage ring gear associated with the first three planet passing modes, animation of a planet carrier revolution can be accessed in supplementary video file.



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
