# Peer review of "Identification of operational deflection shapes of a wind turbine gearbox using fiber-optic strain sensors on a serial production end-of-line test bench"

_Wind Energy Science, 2024_

## Author Comment (AC1)

**Response to Referee Comment #1, wes-2024-83, 16 Sep 2024**

**Unai Gutierrez Santiago, on behalf of the authors**

**October 28, 2024**

**Citation:** https://doi.org/10.5194/wes-2024-83-RC1

We sincerely thank the referee for reviewing our manuscript and providing constructive feedback to improve our manuscript. We have revised the manuscript accordingly. Below are the original comments from RC#1 in black and our responses in blue.

In this paper, the authors present a system to measure the operational deflection shapes (ODSs) of the ring gear of a gearbox using fiber optic strain sensors. The shapes then have application to the measurement of mechanical torque in the gearbox, currently not typically available, for the purposes of remaining useful life estimation or even gearbox condition monitoring. In general, the article is interesting and well-written. I have provided the following comments and some grammatical suggestions at the end.

In the Abstract, gear tooth root strain gauge measurements are described in comparison to the fiber optic ODS and torque measurements. Although I generally understand the comparison, I think most readers might be a bit confused here as Abstract speaks to torque measurement up to this point. It might be simpler just to delete "Compared to conventional gear tooth root strain gauge measurements," as the deflections measured by them relate to the sun-planet and/or planet-ring tooth load distributions rather than the deflection of the ring gear from torque. Nothing is lost from the sentence by deleting this and it is much simpler and direct. Additionally, I might recommend changing "research the deformations caused by gear mesh events" to "measure the deformations caused by planet gear passage events". The primary content measured by the fiber optics is the planet passage, as evidenced by Figure 8, not the gear meshing that occurs at a much higher frequency.

We agree to delete "Compared to conventional gear tooth root strain measurements" as the resulting sentence conveys the desired meaning in a more direct way. We also agree that "planet gear passage event" is more precise than "gear mesh events" because several teeth engage in meshing actions throughout the deformation peaks shown in Figure 8.

Introduction, line 50: A small point, but I recommend deleting "substantial" an just saying "impact on the main frame, tower and foundation". The implication is that tower-top mass (and specially from the drivetrain) substantially impacts these costs - unless a reference can be given, we have found that the rotor loads are the main driver. Reducing drivetrain mass was not a particularly strong driver as described in https://www.sciencedirect.com/science/article/pii/S0306261923006360.

We agree that the use of "substantial" is probably disproportionate. We wanted to convey that torque is the main driver for sizing the drivetrain (line 49), and the weight of the drivetrain has a cascading effect on the main frame, tower, and foundation.

Introduction, line 62: Another minor suggestion - an additional reference might be nice at the end of this paragraph. I recommend https://doi.org/10.5194/wes-7-387-2022.

Citation added, indeed the trend to replace roller element bearings with journal bearings is discussed in the article "Wind turbine drivetrains: state-of-the-art technologies and future development trends" from Nejad et al..

Introduction, line 70: Unmentioned here is that estimating mechanical torque from electrical currents also contains inherent uncertainties in the converter, generator, and gearbox efficiencies. I can't think of a reference that quantifies this, but I believe this to be generally accepted. It would be nice to add this point here. In terms of the impact of load on at least bearing fatigue, since the fatigue is roughly proportional to the cube of the load, then even a 10% error results in a 33% error in fatigue - this even in normal operating conditions.

I will admit I don't know wether this uncertainty or the omission of torque fluctuations in dynamic events is more important, but I wonder.

We agree and have added the sentence "Even in normal operation, relatively large errors are expected when using generator currents because the power losses in the generator and the gearbox vary with torque and other operating conditions and are generally unknown". Currently, we are unable to anticipate which of these two factors, measurement uncertainty or the lack of information from torque fluctuations, is more important.

Introduction, lines 109-111: Initially, when I read the stated third contribution of the paper itl, at least to me, read as the most important contribution. So much so that I was going to suggest it is important enough to merit being "elevated" as part of the Title. After reading the remainder of the article; however, it feels like this bullet point is a bit of an overstatement. Certainly such measurements could be used in a framework, but the framework itself is only mentioned rather than being proposed (and certainly described) in this paper. I'm not quire sure what I'd recommend here - to leave this as a third contribution and "soften" the contribution, or put this in text as the usefulness of such a system and analysis. I simply ask the authors to reconsider how this third bullet is written with respect to how the article itself is written.

This comment aligns with Referee #2, and we agree that the proposed framework to track operation deflection shapes over time for fault detection is more a proposed potential application of the results of the paper rather than an actual contribution of the paper. We have reworded the contributions accordingly. The conclusions section and the abstract have also been slightly modified to emphasize this remark.

Section 2, line 150: Could you add a short mention of what Ab is? I take it that the periodic and structural modes are Aper and Asys, but I don't see mention of what Ab represents.

We have added definitions of the variables $A^{\text{sys}}$, $A^{\text{b}}$ and $A^{\text{per}}$ to the revised manuscript. As this comment aligns with another from Referee #2 we have modified this section with an explicit explanation of how the influence of the unknown input $u(t)$ is modeled inside the extended system matrix $\bar{A}$.

Section 4.1, line 265: A similar comment as before regarding "mesh events versus "planet passage events. Here in this line I believe the phenomenon being described is better represented as "mesh forces as each planet passes the measurement point on the ring gear occur at different times. That is, in Figure 8a, one only really sees 7P content, not 83P content.

We agree and have reworded several sentences in this section to clarify that within each stage, deformations caused by the mesh forces as the planets pass close to the measurement locations occur at different times. Additionally, each stage has a different spacing between deformations because each stage has a different planet carrier rotational frequency.

Section 4.1, line 279: Similar to the Abstract, I recommend "gear mesh events be changed to "planet passage events.

Corrected.

Figure 10: In an earlier figure, red and blue were used as 2 different sensors, but here the two colors are not labeled. Could a legend be added here, or other description? I don't think the colors have the same meaning.

Indeed, blue and red don't relate to the sensor number as in Figure 8. The mode shapes involve all sensors on the different stages, and an explanation has been added to the caption to clarify how the mode shapes are represented. Each shape is defined by two conjugate vectors with a size equal to the number of sensors; the two different colors in the plot represent the real and imaginary parts. Green and orange have been chosen to differentiate from Figure 8. This comment has been applied to the rest of the plots showing mode shapes in the revised manuscript.

Figure 13 and associated text: I'll admit I don't see much point in this figure, but maybe I'm really missing it? Maybe just say that the modules were not found to be time variant. Or maybe the relative magnitudes of each mode could be listed in Table 5 - I think the main point is that mode 2 representing 7P has the highest magnitude, just reinforcing that the signals are primarily comprised of this as Figure 8a already shows.

We agree and have erased this figure from the revised manuscript. As suggested, we have added the average values of state modulus to Table 5.

Minor grammatical comments:

Line 44: I believe the year for "Stehly et al." is missing. It would typically look like "Stehly et al. (2016)" or "Stehly et al. (2021)".

Corrected.

Line 86: A comma is needed here "... or a shaker, EMA relies...". Similarly on line 90: "... with OMA, because in OMA...".

Corrected.

Line 115: I might suggest "described" instead of "shown" here.

Corrected.

Line 158: I believe this inline citation style should be "Verhaegen and Verdult (2007)".

Corrected.

Line 174: I believe "it" is missing and should read, "it has a sustained oscillation".

Corrected to "the system exhibits a sustained oscillation" to avoid confusion.

Line 176: I believe the "i" in "ith" should be italicized.

Corrected.

Line 243: "barking" torque should be "braking" torque.

Corrected.

In the References, Veers et al. 2022 can be updated from https://doi.org/10.5194/wes-2022-32 to https://doi.org/10.5194/wes-8-1071-2023.

Corrected.

Again, we thank the reviewer for the positive feedback.

---

## Author Comment (AC2)

**Response to Referee Comment #2, wes-2024-83, 30 Sep 2024**

**Unai Gutierrez Santiago, on behalf of the authors**

**October 28, 2024**

**Citation:** https://doi.org/10.5194/wes-2024-83-RC2

We sincerely thank the editor and the referee for reviewing our manuscript and providing constructive feedback to improve our manuscript. We have revised the manuscript accordingly. Below are the original comments from RC#2 in black and our corresponding responses in blue.

This study introduced a methodology for input torque estimates based on identification of operational deflection shapes of a wind turbine gearbox using fiber-optic strain sensors and Multivariable Output-Error State-sPace (MOESP) subspace identification. A brief literature was given to motivate the current research. Needed theoretical discussions were provided along with a description of the test gearbox, which is composed of three planetary stages and one parallel stage. Experimental data was collected from an end-of-line test bench and used in the development and assessment of the proposed methodology. The reviewer has some comments as detailed below. The few critical for the authors to address are bolded. The manuscript is recommended to be revised before it can be considered for publication by the journal.

**1 Abstract:**

- line 5: "framework" may be better replaced by "method"; input torque "measurements" may be better expressed as "estimates".

  We have chosen the term "algorithm" instead of "framework" which we believe is more precise and have erased the term "measurement" for clarity. We agree with the term "torque estimate" as already used in line 12 of the abstract.

- line 8: "consistent" might mean agreement with expectations here. Please reword to make it clear.

  We have reworded this sentence for clarity; we meant that the estimated values were consistent regardless of the identification parameters chosen to run the algorithm, see Sections 4.1 and 4.2.

- **line 14: statement on "operational deflection shapes over time" being proposed to enhance condition monitoring appears not substantiated in the manuscript but only briefly discussed as one of future benefits. Please consider removing or rewording.**

  This comment aligns with Referee #1. We agree that this is a potential future application of the method developed in the current article. We have removed the last sentence of the original abstract and made a clearer statement for future research recommendations in the Conclusions section.

**2 Introduction:**

- line 37: please elaborate on "the influence of surface friction."

  This sentence explains the reason behind the trend to increase the hub height of wind turbines, which was mentioned a few sentences before. We are referring to the influence of the ground on the inflow wind conditions and how modern turbines tend to be taller to exploit better incoming wind flow conditions. We have slightly modified the sentence for clarity.

- lines 38-39: in the sentence, "more power ... enhancing reliability", it appears the first two phrases apply to a plant and the next two phrases apply to individual turbines. Please clarify.

  The sentence in lines 38-39 refers to one of the reasons for increasing the power of individual turbines, but in this case, we are referring to the reliability benefits at plant level; We have reorganized and reworded the sentences for clarity.

- line 40: "a substantial cost reduction in costs" appears not having a direct causal relationship with "large rotors", which appears directly leading to improved energy production or revenue.

  We have modified the sentence to make it more clear that one of the motivations to move to larger rotors is the fact that energy production tends to grow faster than the costs and therefore generally larger wind turbines have lower lCoE. The new sentence reads: "The increase in energy captured by the rotor is bigger than the increase in overall turbine costs because blade lengths can be increased while many other costs remain fixed, generally leading to lower LCoE in larger turbines".

- line 68: please double check whether the torque is estimated by using generator currents.

  This point has also been addressed in a comment from RC#1. Estimating mechanical torque from electrical currents is discussed in lines 67-70 as an alternative to direct measurements. In our opinion, this is not an appropriate alternative because it cannot capture torque fluctuations in damaging events, such as an emergency stop where the mechanical brake is applied. We have also added here the uncertainties associated to the unknown power losses in the generator and the gearbox, which vary with torque and other operating conditions.

- lines 75: please consider adding the technology from AeroTorque (https://www.pttech.com/aerotorque-torsional-dampers/).

  We were not aware of this technology. Based on our current understanding, it appears to be a device designed to mitigate peak loads and torsional oscillations. However, it does not seem to provide measurements of these loads. Without in-depth knowledge of this device, we have elected not to include it in the current article. Nevertheless, we will investigate this suggestion further to better understand its capabilities.

- **lines 109-111: the statement appears not substantiated in the manuscript. Please consider removing.**

  We agree and have removed the statement from the explicit contributions of the article as this comment aligns with the comments from Referee #1. We have reworded this statement, out of the contributions, to emphasize the potential applications of torque measurements for consumed life estimation and operational deflection shapes as an indicator for fault detection. The conclusions section has also been slightly modified to emphasize this remark.

**3 Subspace system identification framework:**

- It appears this section is on the methodology not a framework, please rename the section title accordingly.

  We agree and have changed the title to be more precise. The new section title is now "Formulation of subspace system identification method".

- line 121: is "estate" a typo? Please double check.

  Corrected.

- line 127: is "innovation signal" supposed to be "excitation signal"?

  We have chosen a so-called innovation state-space representation described in equations 1 and 2. This can be derived from a general state-space representation with process and measurement noise. The excitation signal is the input signal $u(t)$. The term innovation signal refers to $e(t)$ as defined in line 127. We have modified this section, starting with a general state space representation and explicitly explaining how the influence of the unknown input $u(t)$ is modeled inside the extended system matrix $\bar{A}$. The innovation representation is introduced later for identification purposes.

- line 150: please define variables in Eq. (6).

  We agree that the variables $A^{\mathrm{sys}}$, $A^{\mathrm{b}}$ and $A^{\mathrm{per}}$ were not properly defined and we have added their definition to the revised manuscript.

**4 Experimental setup:**

- **line 222: is the proposed sensing technology supposed to be robust and easier to commercialized?**

  We believe it is, as discussed in the introduction, fiber-optic sensing is already extensively used in other wind turbine components like blades. Fiber-optic sensors based on FBGs have a proven track record and have been commercially available for a long time. Interrogator technology is evolving rapidly, and promising developments may reduce their costs, making them more attractive. Regarding robustness, we are very satisfied with the performance of the sensors. This was one of the key items in a recent extensive field validation campaign, which we wish to publish soon.

**5 Identification of operational deflection shapes:**

- line 317: please add a brief explanation on model validation not considering Kalman filter.

  We have added a sentence to make it more clear how we are using the one-step-ahead predictor defined in Equation 10 without the Kalman filter. Without the Kalman filter the states of the system are defined only by system matrix and the initial conditions. That it why we consider the behavior as an autonomous system oscillating from a non-zero condition. Because the identified damping for all deflection shapes is very small, the modules of all states show very little variation (Figure 13).

- **page 24, Figure 17: please consider adding a comparison against torque estimated using SCADA data.**

  We don't fully understand this comment. In the test bench used for the experiments presented in the article, the test bench SCADA uses the signals from the torque transducers installed between the high-speed shafts of the gearboxes and the electric motors. In Figure 17, we already plot the "Test bench torque sensors" from the torque transducer signals as the mean of the equivalent torque at the low-speed shaft.

**6 Conclusions:**

- **Plese add a brief discussion on the potential impact, in terms of both commercial and R&D, of the presented methodology.**

  We agree and have added a sentence to emphasize the benefits for a fleetwide implementation in commercial wind turbines and also the use during validation of new products.

  Again, we thank the reviewer for the positive feedback.